# Bilateral Atypical Femoral Fractures after Bisphosphonate Treatment for Osteoporosis: A Literature Review

**DOI:** 10.3390/jcm12031038

**Published:** 2023-01-29

**Authors:** SeokJoon Hwang, Minsu Seo, Dongin Lim, Min Suk Choi, Jin-Woo Park, Kiyeun Nam

**Affiliations:** 1Department of Physical Medicine and Rehabilitation, Dongguk University Ilsan Hospital, Dongguk University College of Medicine, Goyang-si 10326, Gyeonggi-do, Republic of Korea; 2Department of Thoracic and Cardiovascular Surgery, Dongguk University Ilsan Hospital, Dongguk University College of Medicine, Goyang-si 10326, Gyeonggi-do, Republic of Korea

**Keywords:** femoral fractures, bisphosphonates, alendronate, osteoporosis

## Abstract

Introduction: This literature review aimed to investigate the incidence, anatomical concerns, etiology, symptoms, diagnostic tools, management, and prognosis of bisphosphonate (BP)—associated bilateral atypical femoral fractures (AFFs). Methods: The PubMed, Cochrane Library, Web of Sciences, and CINAHL databases were searched up to 20 March 2022. All cases of bilateral AFFs were included, excluding those without any bisphosphonate treatment information and those in which the femoral fracture did not precisely fit into the diagnostic criteria for AFF. Results: We identified 43 patients with bilateral AFFs associated with BP use and conducted a comprehensive analysis. Among 43 patients, 29 (67%) had prodromal symptoms. Regarding the simultaneity of fracture, 21 cases (49%) occurred simultaneously, and 22 cases (51%) occurred sequentially. Alendronate was the most commonly used BP treatment (59%). Regardless of the medication type, BP intake duration was more than 5 years in 77%. The initial diagnosis was performed using X-rays in all cases. A total of 53% of patients had complete fractures, and all patients underwent surgical treatment. Among the remaining patients with incomplete fractures, 18% and 29% received surgical and medical treatments, respectively. After BP discontinuation, teriparatide was most commonly used (63%). Conclusions: The careful evaluation of relevant imaging findings in patients with thigh/groin pain allows the identification of early incomplete fractures and timely management. Since the rate of contralateral side fractures is also high, imaging studies should be performed on the asymptomatic contralateral side.

## 1. Introduction

Osteoporosis is characterized by a progressive decrease in bone mass, leading to increased bone fragility [1]. This systemic skeletal disorder disrupts bone microarchitecture and deteriorates bone tissue [2]. Osteoporosis can cause fractures, leading to disability and mortality [3]. Bisphosphonate (BP) is commonly recommended as the first-line treatment for osteoporosis [4]. Since osteoporosis prevalence increases with aging, a larger population might receive long-term BP therapy [5]. Although BP has been proven to decrease the incidence of proximal femoral and fragility fractures in osteoporotic patients, there are known side effects, such as atypical femoral fracture (AFF) [6,7]. The association between long-term BP therapy and AFF was documented in many cases [8]. To investigate the incidence, anatomical concerns, etiology, symptoms, diagnostic tools, management, and prognosis of BP-associated bilateral AFFs, we reviewed published articles in the literature, with the initial presentation of a unique case of bilateral femoral fracture that was misdiagnosed as coronavirus disease of 2019 (COVID-19) vaccination-associated paraplegia.

### Case Report

A 64-year-old woman with a past medical history of hypertension, dyslipidemia, and osteoporosis (with medication) mentioned her suffering from gait abnormality due to discomfort in both thighs a week after COVID-19 vaccination. After suffering for over a month, she visited a local medical orthopedist and underwent lumbar spine magnetic resonance imaging (MRI) and lower extremity radiography. There was no evidence of bony abnormalities, and the patient was advised to undergo tertiary hospital treatment. On physical examination, the range of motion was normal, and the manual muscle test was weak. However, all were above grade 4. Her Modified Barthel Index score was 75, and her Berg Balance Scale score was 46. She was able to stand for more than a minute but was unable to stand on one leg. Both ankle-jerk reflexes were hypoactive. Laboratory data and X-ray findings were normal (Figure 1). The patient was admitted to the Department of Neurology for further evaluation.

A cerebrospinal fluid study was confirmed to be normal. A nerve conduction study, which recorded the extensor digitorum brevis (EDB) muscles, showed no response in both common peroneal nerves. Sensory nerve studies were all within the normal range. F-wave studies were absent in both common peroneal nerves, while others were normal. Due to the hypoactive ankle-jerk reflex, weakness in the lower extremities, abnormal findings in the nerve conduction study, and the onset of symptoms after vaccination, she was diagnosed with a side effect caused by COVID-19 vaccination, and acute neurological disorder had to be ruled out. After receiving conservative treatment, the patient was transferred to the Department of Rehabilitation Medicine for paraplegia and gait disturbance. We initiated a rehabilitation program that included gait therapy. While receiving physical therapy, she complained of greater pain in both thighs with obvious knocking tenderness. Nerve conduction studies and needle electromyography was conducted, and the results were similar to those of her previous study. Ultrasonography revealed bilateral agenesis of the EDB. We performed a bone scan (Figure 2), which showed focal increased uptake in both lateral bony cortices of the proximal femurs. Additionally, thigh and hip MRI (Figure 3) were performed. Focal cortical thickening was observed in both proximal femurs. After taking the patient history in detail again, we confirmed that she had been taking 150 mg of ibandronate once a month for 10 years without a drug holiday. The patient was diagnosed with incomplete AFF of both thighs. BP was immediately stopped, and a prophylactic trochanteric femoral nail was placed bilaterally to prevent future complete fracture of the proximal femur (Figure 4). Long versions of trochanteric nails were used.

## 2. Materials & Methods

We identified and reviewed previously published articles on bilateral AFFs by searching Medline, SCOPUS, Embase, Web of Science, the Cochrane Central Register of Controlled Trials, the World Health Organization International Clinical Trials Registry Platform, and the clinical trial registry and database of the U.S. National Institutes of Health (clinicaltrials.gov accessed on 20 March 2022) on 20 March 2022. We placed no restrictions on language or publication year in our search with the following keywords: bilateral, atypical femur fracture, and bisphosphonate.

We included all cases of bilateral AFFs, regardless of the duration or type of BP intake. We excluded (1) cases with unilateral fracture, (2) those without any BP treatment information, and (3) those in which the femoral fracture did not precisely fit into the diagnostic criteria for AFF from the ASBMR [9].

## 3. Result

### 3.1. Description of the Included Studies

The search strategy outlined above was used to identify the articles. One hundred-ninety-seven articles were identified by searching the databases. After screening for duplicate records, 84 studies were excluded. After screening titles and abstracts, 65 articles were excluded. Therefore, a total of 48 studies reporting 55 patients were analyzed. Among these, 13 cases were excluded because 6 cases did not provide sufficient information. Five patients had atypical unilateral fractures. One patient did not receive BP treatment for osteoporosis, and the other reported an AFF in the femoral neck, which met the exclusion criteria of AFF. Finally, a total of 43 patients, including the case report of our patient, were analyzed in this systematic review (Figure 5). We identified 43 patients with bilateral AFFs who received BP for osteoporosis in 39 studies (Table 1).

### 3.2. Epidemiology and Clinical Manifestations

The average patient age was 68.8 (range: 36–83) years. The patient age distribution was 2 cases (5%) under 50 years, 12 cases (28%) between 50 and 65 years, 17 cases (40%) between 66 and 75 years, and 12 cases (28%) over 76 years. Of the 43 patients, 11 (26%) had an antecedent disease. A total of 2 cases [21,38] had multiple myeloma, and 9 cases [10,20,21,22,24,28,29,35,44] were related to the use of glucocorticoids and steroids. Only 2 patients (5%) [11,20] were men, and one of them was using glucocorticoids. The remaining 41 patients (95%) were women. Table 2 summarizes the overall characteristics of patients with BP-related bilateral AFFs.

The main complaint was pain in 26 patients (60%). The remaining 17 patients (40%) visited the hospital due to minor trauma. Of the 17 patients, 3 [15,31,44] had prodromal symptoms, and 14 [10,11,12,13,18,20,22,23,29,35,36,37,40] did not have any symptoms before the trauma. Prodromal symptoms included thigh or groin pain in all patients. Two patients [41] complained of an inability to walk in addition to pain. The period from symptom onset to hospital visit in 29 patients with prodromal symptoms was within a day in one case (5%) [33], within a month in 3 cases (15%) [8,26,41], between a month and a year in 9 cases (45%) [24,25,28,32,38,42,43,44], and over a year in 7 cases (35%) [1,14,27,34,39,45]. Details of the remaining 9 cases were unknown.

Among 43 cases of bilateral AFFs, 21 (49%) occurred simultaneously, and 22 (51%) occurred sequentially. In 3 cases [29,32,39] of simultaneous bilateral AFFs, patients complained of unilateral hip pain. The contralateral side was completely asymptomatic. In the 22 sequential cases, 11 cases (50%) occurred within a year, 7 cases (32%) occurred between 1 and 3 years, and 4 cases (18%) occurred more than 3 years after the first unilateral fracture. The longest duration after the first unilateral fracture was 4 years.

We classified the 86 fracture sites into 3 categories according to the anatomical site (divided by one-third from the proximal fracture line under the lesser trochanter to the distal fracture line above the femoral condyles). A total of 40 fractures (47%) were in the near third, 44 fractures (51%) were in the middle third, and the remaining 2 fractures (2%) were in the far third.

### 3.3. Diagnosis

The initial assessment was performed using a simple radiograph in all cases. Confirmative diagnostic modalities were bone scan (26%) [1,8,16,19,24,25,26,28,30,45], MRI (14%) [1,16,19,39,46], and computed tomography (CT) (9%) [1,16,25,28]. In 2 cases (5%) [14,32], patients visited the hospital with thigh pain, but the diagnosis was not made. Eventually, they fell and underwent surgery. A single case (2%) [16] was misdiagnosed as metastatic bone disease from primary breast cancer. Our case was misdiagnosed as an acute neurological disorder caused by COVID-19 vaccination.

### 3.4. BPs

In more than half of the cases (23 cases, 59%), patients were administered alendronate alone. Alendronate intake duration was over 5 years in 14 cases (66%), under 5 years in 7 cases (33%), and unknown in 2 cases. Risedronate alone was used in 4 cases (10%), all with intake duration over 5 years. Two patients (5%) had been taking zoledronic acid alone for 5 and 10 years, respectively. Two patients (5%) had been taking ibandronate for 10 years, and one patient was taking minodronic acid for 3 years. Seven patients (18%) had a history of taking ≥2 BPs. Among these cases, 6 took more than 5 years of BPs in total. In 4 cases, we could not identify the BP type.

According to BP intake duration, regardless of the medication type, 30 cases (77%) received more than 5 years of BPs. Additionally, 3 cases (8%) took BPs for less than 3 years, and 6 cases (15%) received BPs for 3–5 years. The shortest duration of BP-induced AFF was 20 months, with 35 mg of alendronate weekly [10].

Ten AFF cases (23%) occurred even after BP was discontinued. Among these cases, 2 [39,42] had changed BP to denosumab 2 years and 3 years ago, respectively. In 5 cases [10,13,31,33,40], while BP was stopped after unilateral AFF, a contralateral fracture occurred. Three cases [23,34,35] were on drug holidays for 6 months, 1 year, and even a decade, respectively.

### 3.5. Operative Management

When the 86 fractures were classified according to their degree, 46 (53%) were complete fractures, and 40 (47%) were incomplete fractures. All 46 complete fractures underwent surgery. Intramedullary nailing was performed in 41 cases (91%). The remaining patients underwent plate-screw fixation. Of the 40 incomplete fractures, 14 (35%) underwent prophylactic intramedullary nailing, and only one case [8] underwent plate-screw fixation. A total of 56 femurs underwent intramedullary nailing, of which 32 (57%) were proximal fractures and 24 (43%) were middle fractures. Among the 32 proximal fractures, 23 (88%) femurs had nailing with a cephalomedullary component, and the remaining had standard femoral nailing. For the 24 middle fractures, only 6 (25%) femurs underwent nailing with a cephalomedullary component. The remaining 18 (75%) femurs underwent standard femoral nailing without trochanteric nails. Medical treatment was administered to 25 women (63%). In total, 61 femurs (71%) were treated surgically, and 25 femurs (29%) were treated medically.

### 3.6. Osteoporosis Treatment after BP Discontinuation

All patients ceased BP intake after bilateral AFFs. However, 10 patients [12,18,20,21,22,24,25,28,34,37] continued to take BP even after surgery for unilateral AFF due to an inappropriate diagnosis, which eventually resulted in bilateral fractures. In only one case [31], BP was continued despite a unilateral AFF diagnosis.

We identified follow-up osteoporosis prevention drugs after AFF diagnosis in 24 patients. The drug that was most frequently taken after BP discontinuation was teriparatide (15 cases, 63%). Seven patients took teriparatide with calcium and vitamin D. Five patients (21%) switched from BP to calcium and vitamin D without teriparatide. Two patients (8%) stopped BP and did not take any medication for osteoporosis. In one case, BP was changed to denosumab. In another case, BP was changed to strontium ranelate.

### 3.7. Complications

Among 61 surgically treated femurs, 8 (13%) showed a poor prognosis. An incomplete union was observed in 7 femurs, and one had an infection. Reoperation was performed in 6 femurs, and the remaining 2 femurs [40] were observed because patients refused reoperation. The period between operation and reoperation was within a year in 4 cases, with the longest term of 5 years. Reoperation involved replacing the intramedullary nail and using autogenous iliac bone grafts in 3 cases (50%) [10,22,31]. Plate-screw fixation was performed in 2 cases [12,34], and the remaining case [35] underwent intramedullary nailing once again.

Among the 25 medically treated femurs, 10 (40%) showed a poor prognosis, which eventually led to surgical treatment. In 6 femurs (60%) [35,39,41,45], aggravation was shown clinically or radiologically, and prophylactic intramedullary nailing was performed. In 4 femurs (40%) [26,43], patients experienced minor trauma, and a complete fracture occurred. Intramedullary nailing (3 cases) and plate-screw fixation (1 case) were performed.

In successfully treated patients, the complete bone union period ranged from 3 months to 5 years (average duration: 12.7 months). The average bone union period was 10.1 months in surgically treated patients and 20.4 months in medically treated patients. Bone union duration was presented in only one of the 6 reoperation cases, with a duration of 24 months.

## 4. Discussion

### 4.1. Epidemiology

In the literature, there is a strong correlation between BP intake duration and AFF [7,9,47,48]. The incidence in the first year of alendronate use is 2 per 100,000 patient-years, which increases to 25 per 100,000 patient-years after 3–5 years of treatment and to 130 per 100,000 patient-years with more than 8 years of treatment [49]. In 2013, the American Society for Bone and Mineral Research (ASBMR) published the definition of AFFs, in which 4 major features, regardless of the presence of minor features, are required (Table 3) [9]. As stated by ASBMR, AFF can be observed bilaterally [9]. The risk of developing AFF on the contralateral side should be known even if the fracture is not bilateral. Therefore, at the time of the initial admission, the contralateral femurs of every patient were evaluated. Even if no complete fracture is visible on radiological imaging, stress reactions, which may be considered precursors of fracture, can be detected, and preventive measures can be taken [50]. In a case-control study, 28% of patients with AFFs had a contralateral fracture, as compared to a 0.9% contralateral fracture rate in patients with a typical fracture pattern (odds ratio: 42.6, 95% confidence interval: 12.8–142.4) [7]. Contralateral fractures occurred from 1 month to 4 years after the index AFF, with an average time of 10.3 months [51]. Probyn et al. also reported that contralateral AFFs were diagnosed within 12 months of the index fracture in 76.9% of the cases and within 3 years in 88.5% of the cases [52]. In our study, 49% of patients with a first fracture were immediately diagnosed with an incomplete fracture on the asymptomatic contralateral side. Half of the remaining 51% were diagnosed within 1 year and up to 4 years later, which is similar to the trend observed in previous studies.

In our study, AFF occurred in 77% of patients with BP intake for more than 5 years. On the other hand, AFF occurred in 23% of patients with less than 5 years of BP intake, with the shortest occurrence of 20 months. Therefore, while the correlation between BP exposure duration and AFF occurrence is clear, it is necessary to recognize the possibility of AFF even within 5 years. In addition to BP treatment duration, there are several other noteworthy risk factors for AFFs, including Asian ancestry, shorter height, higher weight, and glucocorticoid use for one year or more [49]. In our study, glucocorticoid use was confirmed in 9 of 43 patients. AFF can be brought on by medical disorders other than osteoporosis, including collagen diseases, chronic pulmonary diseases, rheumatoid arthritis, asthma, and diabetes [53]. Despite being significant risk factors for hip and other osteoporotic fractures, older age, prior fractures, and lower bone mineral density do not significantly enhance the risk of atypical fractures [49].

### 4.2. Pharmacology and Usage of BP

For more than two decades, BPs have been the most frequently utilized osteoporosis medication. BPs are antiresorptives that reduce bone turnover markers to low premenopausal concentrations and reduce fracture rates (vertebral by 50–70%, non-vertebral by 20–30%, and hip by approximately 40%) [54]. BPs bind strongly to bone minerals, having an offset of the antiresorptive effect. This effect is measured over months to years. Zoledronate has the highest binding affinity to bone, followed by alendronate, ibandronate, and risedronate [55]. The optimal BP treatment duration remains unclear. Long-term (usually 5 years) continuous use of oral BPs is usually interspersed with drug holidays of 1–2 years to minimize the risk of AFFs [54]. Shorter drug holidays (6–12 months) would be preferable due to the faster antiresorptive effect mitigated by risedronate. The American Association of Clinical Endocrinologists guidelines suggested “drug holidays” after 5 years in moderate-risk patients and after 10 years of BP therapy in high-risk patients [56]. Patients at a high risk of fracture correspond to those with a low bone mineral density ≤ −2.5, sustaining a previous vertebral fracture, older than 70 years, exhibiting high bone turnover markers, or undergoing glucocorticoid therapy [57]. They may benefit from commencing treatment with a selective estrogen receptor modulator, such as teriparatide, if BP therapy has been stopped after 5 years [58]. In our study, 3 cases of AFF occurred during drug holidays. Contralateral fractures occurred in 5 cases, although BP was stopped after unilateral AFF. Therefore, even after BP discontinuation, the continuous monitoring of the fracture risk assessment tool score and bone turnover markers is necessary to guide treatment selection.

### 4.3. Diagnosis

Clinical features are most important in the early diagnosis of AFF. Approximately 32–76% of patients who have incomplete or developing AFF present with a prodrome of groin or hip pain [9,59]. In our study, 67% of patients had prodromal symptoms, such as hip or thigh pain. Prodromal pain, which manifests as pain in the anterior or lateral thigh or groin, can develop anywhere between two weeks and many years prior to the fracture [60]. Our study confirmed that it took more than 1 year from prodromal symptoms to hospital visits in 35% of the cases, indicating that the pain level before the complete fracture was not severe. Another prodromal symptom may be gait disturbance accompanied by slight discomfort of the femur, as observed in our case report. As a result, any minor prodromal pain in a patient receiving antiresorptive medication should prompt the doctor to take a hip radiograph and check for contralateral symptoms and fractures.

Radiography is usually the initial step and should include a frontal view of the pelvis and two views of the full length of each femur [60]. In 2015, Probyn et al. studied X-ray symmetry in bilateral AFFs [52]. In 26 cases when contralateral X-rays were performed at the time of first admission, signs of localized periosteal or endosteal thickening were found in 92.3% of cases. Locations of the bilateral AFF had a high association, with 76.3% occurring within a distance <5 cm and 53.9% occurring within a distance ≤2.5 cm [52]. The bilateral AFF pairings demonstrated strong agreement for the position of the medial spike and the fracture direction, and their femoral angles were moderately correlated. In conclusion, due to the fact that bilateral AFFs frequently share imaging characteristics, such as location along the femur, it is worth paying attention to the symmetry of the contralateral side [52]. If radiography is inconclusive, a bone scan or MRI should be performed. The most sensitive and specific radiographic findings are the linear cortex transverse fracture pattern and focal lateral cortical thickening [61,62]. In our study, the incidence of bilateral AFF that occurred simultaneously was <50%. More than half of bilateral AFF occurred sequentially, indicating that the initial evaluation of both sides is important. Two patients missed contralateral side screening despite an initial ipsilateral fracture [14,32]. The contralateral side should also undergo radiographic study due to the possibility of fracture. Due to its low sensitivity in early stress fractures, CT is not recommended in the diagnosis of atypical fractures [60]. Active bone turnover can be seen by bone scan using a gamma camera and technetium 99 m-labeled methylene diphosphonate. Thus, stress fractures and AFFs can be easily detected in the third (delayed) phase of bone scan [63]. Bone scanning has great sensitivity, but its lack of spatial resolution limits its specificity [64]. AFFs have MRI features comparable to other stress fractures, with the exception that the pattern is lateral to medial rather than medial [65]. The earliest findings included a periosteal reaction of the lateral cortex with a normal marrow signal. Patients with known AFFs may benefit especially from MRI for contralateral leg screening [60]. Contrast enhancement is not required. In our case, bilateral AFF was confirmed by a bone scan and MRI, as it was unclear on radiography.

### 4.4. Management

#### 4.4.1. Medical Management

In BP users with radiographic evidence of AFF, BP should be stopped because continuation may lead to the worsening of the existing AFF or the development of a new contralateral AFF [9]. According to our study, AFF was not diagnosed in most patients with asymptomatic AFF, who developed a complete fracture from no or minor trauma. The most frequent mechanism of injury seems to be a fall from a standing position or even a routine action such as walking or stepping off a curb. Eventually, BP was used continuously. Of the 22 cases with sequential fractures, only 4 stopped BP after the first fracture. The remaining 18 patients were eventually diagnosed with AFF after a contralateral fracture, and BP was discontinued. Therefore, it is necessary to examine the history of BP use in patients with osteoporosis. It is important to examine whether there has been an impact large enough to cause a fracture or whether the characteristics of the fracture are indicative of AFF. After BP discontinuation, teriparatide is the representative treatment for osteoporosis. Teriparatide targets the parathyroid hormone-1 receptor and is administered via daily subcutaneous injection (20 μg/day) for up to 2 years. Teriparatide, which has been linked to improved bone fracture healing, may be used either alone or as an adjuvant therapy to surgical fixation to increase AFF healing. A systematic review published in 2015 endorsed the use of teriparatide for positive outcomes in AFF [59]. Additionally, a 10-patient series showed that incomplete fractures without radiolucent lines responded to teriparatide alone, whereas those with radiolucent lines required intramedullary nailing. These results suggested that teriparatide worked best for stable fracture sites, either inherently or with surgical fixation [59]. Because of concerns of osteosarcoma, teriparatide treatment is typically limited to 24 months in most countries.

Although a major clinical dilemma is the follow-up management course after a full 2-year treatment with teriparatide, subsequent therapy may be administered with raloxifene (or hormone replacement therapy) in women and in those with bilateral surgical fixation of AFF, denosumab or BPs [66]. However, BP continuation could increase the risk of atypical fractures at skeletal sites other than the femur [67,68,69,70,71,72,73,74].

Teriparatide may be stopped without additional antiresorptive treatment in those who are deemed to be at low risk of osteoporotic fractures or those who have low bone turnover markers after teriparatide. Even so, it is advised to closely monitor bone mineral density and bone turnover markers [66].

Treatment of AFF varies depending on the patient’s severity of pain and the degree of the fracture. First, incomplete AFF without pain can be followed conservatively [75]. In addition to the discontinuation of antiresorptive therapy, patients should refrain from high-impact and repetitive-impact exercises such as jogging or leaping. Patients should start safe weight-bearing activity as soon as pain appears. The treatment of patients with incomplete AFFs and pain is controversial. Successful regimens include a thorough metabolic bone workup, anabolic bone agents, calcium and vitamin D supplementation, and 2-3 months of protected weight-bearing exercise [60]. According to some authors, conservative treatment yields poor results, with only a few patients experiencing pain relief or showing signs of full recovery [76,77]. Additionally, safe weight bearing of both legs may not be practicable if an incomplete fracture is discovered in the opposite femur. Patients with incomplete fractures should be monitored regularly using physical examinations and radiography. Surgery is required if the pain worsens, the fracture progresses, or it does not heal within 2–3 months. In our study, 40% of patients with incomplete AFF who underwent conservative treatment eventually underwent surgical treatment. Prophylactic surgery is also advised for incomplete AFFs, particularly those with severe pain, extensive cortical abnormalities and/or marrow edema on MRI, which are predisposed to delayed or non-union or to proceed to complete AFFs without surgical intervention [77,78].

#### 4.4.2. Surgical Management

Orthopedic care for AFF depends on whether the patient experiences pain and whether the fracture is incomplete or complete. These are challenging fractures to manage because of their complex displacement patterns, altered bone geometry, sluggish healing in the elderly, and the risk of fracture in the opposite limb. These considerations raise questions about recommending protected weight-bearing exercises. Furthermore, AFFs are often associated with increased anterolateral bowing of the femur, which makes intramedullary nail insertion difficult. This can result in intraoperative complications, such as malunion from excessive straightening of the femur (which might cause leg length discrepancy), iatrogenic fracture during prophylactic nailing, and gapping of the fracture site, especially on the medial side. Intramedullary nailing is the first-line treatment for complete AFFs, although the risk of delayed healing and revision surgery may be somewhat higher than that for typical femoral fractures [79]. If bone deformities make it difficult to insert an intramedullary nail, a long locking plate can be employed. However, nails are favored because they can be helpful in difficult-to-heal fractures by allowing the formation of an endochondral callus [60]. A well-founded review of the method of intramedullary nailing according to the location of the fracture in AFF has not yet been published. In our study, intramedullary staining was performed in 92% of cases, and plate-screw fixation was performed in only 8% of cases.

In 2020, Ebrahimpour et al. reported that the complication rate of surgical treatment of AFF was 17.52% [78]. The most frequent reason was non-union, followed by implant failure [78]. The authors argued that AFF might have more postoperative complications and should be operated on by surgeons with more experience. A recent study by Carlo et al. analyzed the efficacy of plate augmentation in non-union after intramedullary nailing in femoral shaft fractures [80]. The study suggested that the patients treated with plate augmentation showed a good consolidation rate in femoral shaft non-union, with good functional recovery and a low complication rate. In our study, 8 of 61 surgically treated femurs (13.1%) showed complications. Most of them resulted from non-union. Additionally, one case resulted from infection at the surgical site [35]. All patients underwent revision surgery, except for a case of surgery refusal [40]. In most cases, revision surgery was intramedullary nailing with an allogeneic bone graft [10,22,31].

### 4.5. Limitation of This Review

Our study had some limitations. The review included only a few patients, mostly from case reports or small cohort studies. A quantitative evaluation of the characteristics, risk factors, and management of bilateral AFFs was not possible due to the lack of a comparison group and the small number of included studies. Increased identification and management of AFFs would arise from greater knowledge of these issues. Therefore, further studies with a higher level of evidence and larger randomized clinical trials are needed to investigate various aspects of bilateral AFFs. Additionally, more data regarding the success rate of conservative management of incomplete fractures, including the use of teriparatide, are required. Finally, future studies are needed to determine the optimal surgical technique to reduce postoperative complications and avoid potential revision surgery.

## 5. Conclusions

This review summarized the key findings on the epidemiology, anatomical concerns, etiology, symptoms, diagnostic tools, management, and prognosis of BP-associated bilateral AFF. Despite its very low incidence, atypical fracture is the most common complication associated with BP, the main drug available for osteoporosis prevention. Prodromal symptoms are common and important in patients receiving BPs. Patients on BP therapy should be monitored regularly for symptoms such as groin pain, knocking tenderness, and painful gait disturbance. Careful evaluation of pertinent imaging findings in patients with thigh/groin pain enables the early identification of incomplete fractures and timely management. Since the rate of contralateral fractures is also high, imaging studies should be performed on the opposite side. The early detection of asymptomatic contralateral AFFs can avoid complete fractures and allow rapid recovery through timely treatment.

## Figures and Tables

**Figure 1 jcm-12-01038-f001:**
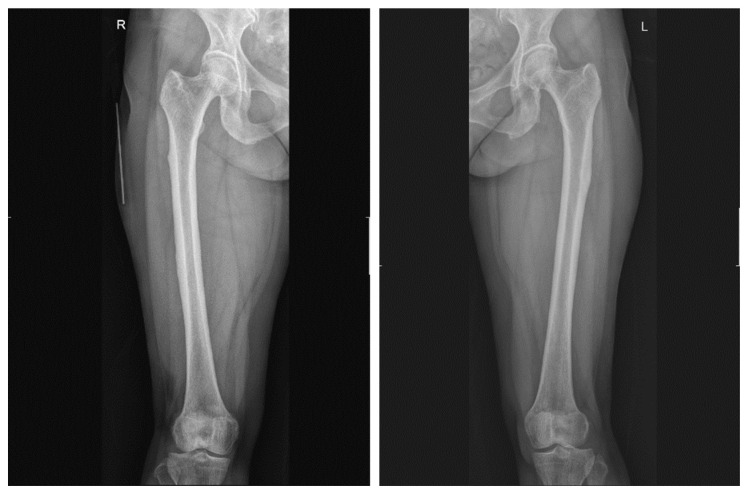
Anteroposterior X-ray of both femurs taken initially.

**Figure 2 jcm-12-01038-f002:**
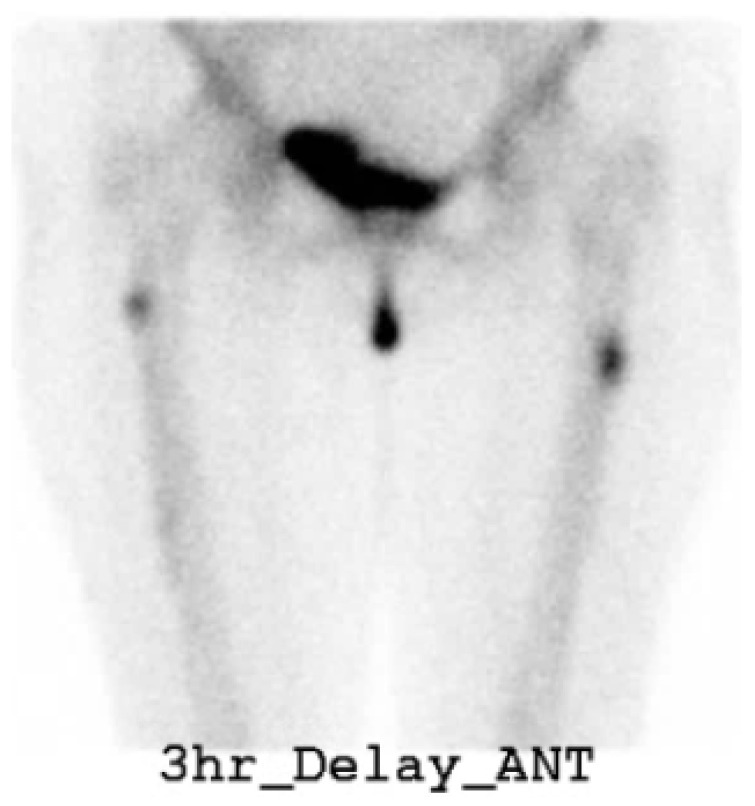
Bone scan (three phases) displayed focal increased uptakes in both proximal femur lateral bony cortex.

**Figure 3 jcm-12-01038-f003:**
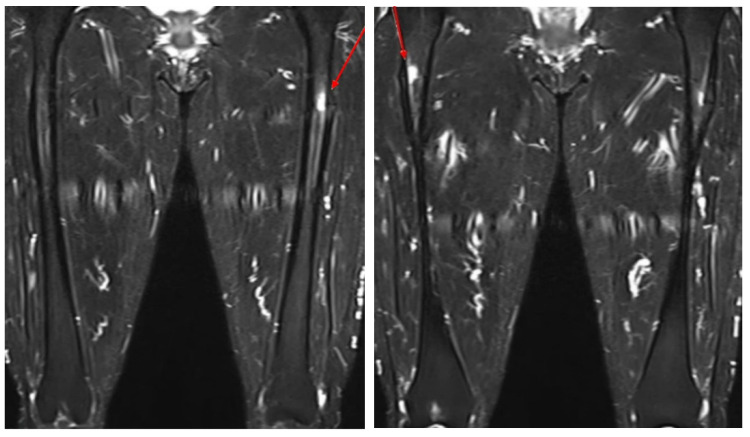
T2 MRI of both hips. The red arrow shows an impending AFF pattern.

**Figure 4 jcm-12-01038-f004:**
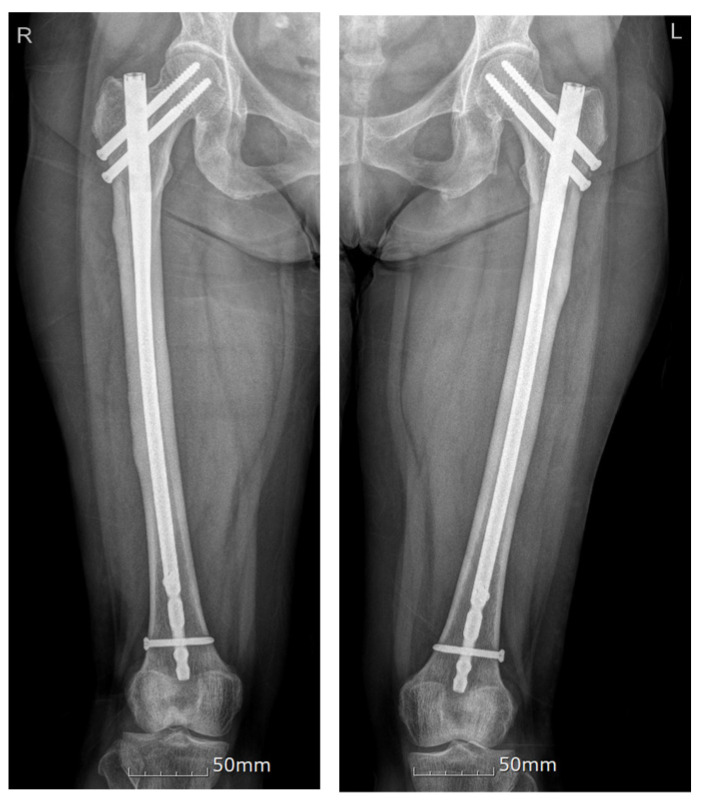
X-ray taken after prophylactic trochanteric femoral nailing.

**Figure 5 jcm-12-01038-f005:**
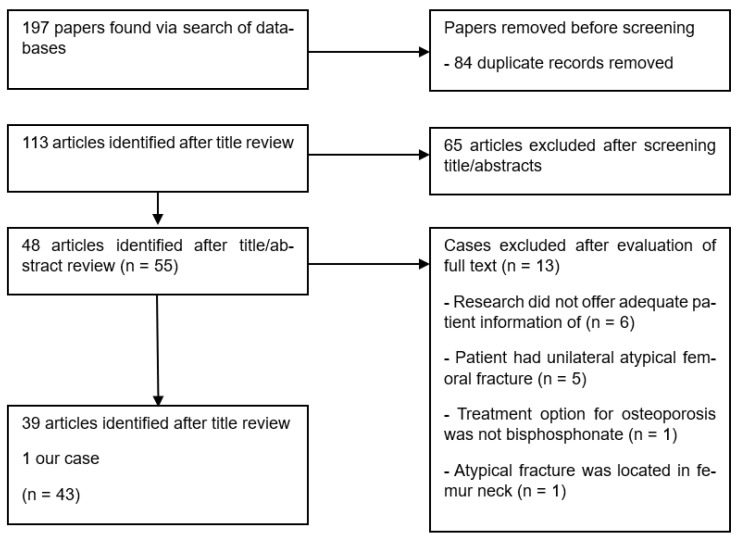
Flow of studies through the review.

**Table 1 jcm-12-01038-t001:** Data of 43 patients with bilateral AFF treated with BP for osteoporosis in 39 studies.

Case	References	Patient (Sex, Age)	Prodromal Symptoms (R/L)	Simultaneity of Fractures	Type and Period of Bisphosphonate	Fracture Sites (R/L)	Surgical Intervention (R/L)	Drugs Changed after Stopping BP	Complete Bone Union Period (R/L)
1	Our case	F, 64	O/O	Simultaneous	Ibandronate 150 mg monthly for 10 years	Prox/prox	PIM nailing/PIM nailing	Teriparatide	NA
2	Ken Iwata et al., 2017 [10]	F, 68	X/X	Sequential (1.5 years)	Alendronate 35 mg, weekly for 20 months	Prox/prox	IM nailing/PIM nailing	NA	12 months/NA
3	Oluwatobi O Onafowokan, 2021 [11]	M, 69	X/X	Simultaneous	Risedronate for 11 years	Mid/mid	IM nailing/IM nailing	NA	NA
4	Raul Parron Cambero et al., 2012 [12]	F, 72	X/X	Sequential (9 months)	Unknown BP for 8 years	Prox/mid	IM nailing /plate-screw fixation	Teriparatide	NA/4 months
5	Eric D. Van Baarsel et al., 2019 [1]	F, 63	O/O	Simultaneous	Alendronate for unknown period	Prox/prox	PIM nailing/PIM nailing	NA	NA
6	Cleto-Zepeda G. et al., 2019 [13]	F, 65	X/X	Sequential (5 months)	Risedronate 35 mg weekly for 7 years	Mid/mid	IM nailing/X	Teriparatide +calcium, vit D	3 months/NA
7	Alessandro Moghnie et al., 2016 [14]	F, 76	O/O	Sequential (2 years)	Alendronate 70 mg weekly for more than 5 years	Mid/mid	IM nailing/IM nailing	NA	NA
8	Tarun Pankaj Jain et al., 2021 [15]	F, 77	O/O	Sequential (5 months)	Alendronate for 7 years	Mid/mid	IM nailing/IM nailing	NA	NA
9	Denise M. van de Laarschot et al., 2016 [16]	F, 50	O/O	Sequential (5 months)	Alendronate 70 mg weekly for 1 year Zoledronic acid 4 mg monthly for 8 months	Prox/mid	X/X	Strontium renelate	NA
10	Venthan Jeyaratnam Mailoo et al., 2019 [17]	F, 71	O/O	Simultaneous	Ibandronate 150 mg monthly for 10 years	Mid/prox	X/X	Teriparatide +calcium, vit D	24 months/ 24 months
11	Joon Kiong Lee, 2009 [18]	F, 82	X/X	Sequential (4 years)	Alendronate 70 mg weekly for 8 years	Mid/mid	Plate-screw fixation /plate-screw fixation	NA	NA
12	Scott M. Sandilands, DO et al., 2016 [19]	F, 77	O/O	Sequential (6 weeks)	Alendronate for unknown period	Prox/mid	IM nailing/PIM nailing	NA	12 months/NA
13	Manuel Roman et al., 2015 [20]	M, 72	X/X	Sequential (2 years)	Alendronate for 11 years	Mid/mid	IM nailing/IM nailing	Teriparatide	5 months/ 7 months
14	Ichiro Tonogal et al., 2014 [21]	F, 53	O/O	Sequential (2.5 years)	Incadronate 10 mg monthly for 5.5 years Zoledronate 4 mg q3monthly for 5 years	Prox/prox	IM nailing/IM nailing	NA	4 months/NA
15	Naoki Kondo et al., 2015 [22]	F, 36	X/X	Sequential (4 months)	Alendronate 5 mg daily for 3.5 years	Prox/prox	IM nailing/IM nailing	Calcium + vit D	24 months/ 12 months
16	Matthew D. Smith et al., 2021 [23]	F, 70	X/X	Sequential (4 months)	Alendronate for 10 years	Mid/mid	IM nailing/X	Teriparatide	11 months/NA
17	Matthijs P. Somford et al., 2009 [24]	F, 76	O/O	Sequential (1 years)	Alendronate 10 mg daily + Alendronate 70 mg weekly for 8 years	Prox/prox	IM nailing/IM nailing	NA	NA
18	Franciso Jose Tarazona-Santabalbina et al., 2013 [25]	F, 73	O/O	Sequential (1 months)	Alendronate 10 mg daily for 7 years Alendronate 70 mg weekly for 6 years	Mid/mid	X/IM nailing	Teriparatide calcium + vit D	12 months/NA
19	Ken Lee Puah et al., 2011 [26]	F, 64	O/O	Simultaneous	Alendronate for 1 year	Dist/prox	X/X	NA	NA
20	Raju Vaishya et al., 2013 [27]	F, 63	O/O	Simultaneous	Alendronate 70 mg weekly for 3 years	Prox/prox	X/X	Teriparatide	NA
21	Indunil Gunawardena et al., 2011 [28]	F, 67	O/O	Sequential (2 years)	Alendronate for 4 years	Prox/prox	IM nailing/IM nailing	Calcium + vit D	NA
22	Yil Ryun Jo et al., 2013 [8]	F, 75	O/O	Simultaneous	Alendronate 70 mg weekly for 8 years	Mid/mid	Plate-screw fixation /PIM nailing	None	NA
23	Kosuke Hamahashi et al., 2020 [29]	F, 57	O/O	Simultaneous	Zoledronic acid for 10 years	Prox/prox	IM nailing/PIM nailing	NA	24 months/ 15 months
24		F, 57	X/X	Sequential (3 years)	Alendronate for more than 10 years	Prox/prox	IM nailing/IM nailing	NA	25 months/ 58 months
25		F, 41	X/X	Simultaneous	Minodronic acid for 3 years	Prox/prox	IM nailing/PIM nailing	NA	19 months/NA
26	Jo Eun Kim et al., 2015 [30]	F, 82	O/O	Sequential (2 years)	Unknown BP for 7 years	Prox/prox	PIM nailing/X	NA	NA
27	H.-y. Zhang et al., 2019 [31]	F, 71	O/O	Sequential (3 years)	Alendronate 70 mg weekly for 5.5 years Zoledronic acid 5 mg yearly for 4 years	Prox/prox	IM nailing/IM nailing	Teriparatide calcium + vit D	NA
28	Fahad Alshahrani et al., 2012 [32]	F, 72	O/X	Simultaneous	Etidronate for 5 years Alendronate 70 mg weekly for 4 years	Prox/prox	IM nailing/X	Teriparatide calcium + vit D	3 months/NA
29	Vivek Sodhai et al., 2019 [33]	F, 73	O/O	Sequential (5 months)	Unknown BP for unknown period	Mid/mid	IM nailing/PIM nailing	NA	12 months/NA
30	Mehmet Okcu et al., 2021 [34]	F, 60	O/O	Simultaneous	Alendronate 70 mg weekly for 3 years	Mid/mid	X/X	NA	NA
31		F, 83	O/O	Sequential (5 months)	Alendronate 70 mg weekly for 4 years	Mid/mid	Plate-screw fixation /IM nailing	NA	NA
32	Mohammad Golsorkhtabaramiri et al., 2020 [35]	F, 78	X/X	Simultaneous	Risedronate 35 mg weekly for 9 years	Prox/prox	IM nailing/X	Calcium + vit D	NA
33	Tatsuki Kobayashi et al., 2021 [36]	F, 80	X/X	Simultaneous	Risedronate 2.5 mg daily for 9 years	Mid/mid	IM nailing/IM nailing	NA	8 months/ 8 months
34	Edelissa Payumo et al., 2018 [37]	F, 75	X/X	Sequential (3 years)	Alendronate for 7 years	Prox/mid	IM nailing/IM nailing	Denosumab	NA
35	Edwin Chiu et al., 2020 [38]	F, 73	O/O	Simultaneous	Zoledronic acid 4 mg monthly for 5 years	Prox/prox	PIM nailing/PIM nailing	NA	NA
36	Elisabetta Neri et al., 2021 [39]	F, 75	X/O	Simultaneous	Risedronate 35 mg weekly for 2 years Ibandronate 35 mg weekly for 5 years	Mid/mid	X/PIM nailing	Teriparatide	NA/12 months
37	Young Sung Kim et al., 2015 [40]	F, 76	X/X	Sequential (1 month)	Unknown BP for unknown period	Mid/mid	IM nailing/IM nailing	None	NA
38	S. K. Ramchand et al., 2015 [41]	F, 82	O/O	Simultaneous	Alendronate for 6 years Risedronate for 1 year	Mid/mid	X/X	Teriparatide	NA
39	J. Selga et al., 2015 [42]	F, 62	O/O	Simultaneous	Alendronate for 8 years Risedronate for 2 years Alendronate for 2 years	Prox/mid	IM nailing/IM nailing	Teriparatide	3 months/ 3 months
40	C.P. Zafeiris et al., 2012 [43]	F, 76	X/O	Simultaneous	Alendronate for 11 years	Mid/dist	X/X	Calcium + vit D	NA
41	Mark Higgins et al., 2016 [44]	F, 71	O/O	Simultaneous	Alendronate 70 mg weekly for 8 years	Mid/prox	IM nailing/IM nailing	Calcium + vit D	NA
42	Morgane Righetti et al., 2017 [45]	F, 67	O/O	Simultaneous	Alendronate for 10 years	Mid/mid	X/X	Teriparatide	NA
43	Stephen J. et al., 2011 [46]	F, 63	O/O	Simultaneous	Alendronate 10 mg daily for 6 years Alendronate 70 mg weekly for 7 years	Mid/mid	X/X	Teriparatide calcium + vit D	21 months/ 21 months

NA, non-available information; R, right; L, left; BP, bisphosphonate; prox, proximal; mid, middle; dist, distal; PIM, prophylactic intramedullary; IM, intramedullary; vit, vitamin.

**Table 2 jcm-12-01038-t002:** Overall characteristics of patients with BP-related bilateral AFF.

	Patient	Total 43
Patients	Age (mean ± SD)	68.8 (±10.2)
Demographics	Gender (female)	41 (95%)
Bisphosphonate use (years)	Alendronate	23 (7.1)
	Risedronate	4 (9.0)
	Ibandronate	2 (10.0)
	Zoledronic acid	2 (7.5)
	Minodronic acid	1 (3.0)
	Two or more bisphosphonates	7 (8.4)
	Unknown	4 (7.5)
Treatment option for AFF (total 86 femurs)	Surgical treatment	61 (71%)
	Non-surgical treatment	25 (29%)
Medication after stopping BP	Teriparatide	15
	Calcium and vitamin D	5
	Denosumab	1
	Strontium ranelate	1
	No medication	2
	Unknown	19

**Table 3 jcm-12-01038-t003:** ASBMR task force 2013 revised case definition of AFFs.

To satisfy the case definition of AFF, the fracture must be located along the femoral diaphysis from just distal to the lesser trochanter to just proximal to the supracondylar flare. In addition, at least four of the five major features must be present. None of the minor features are required but have sometimes been associated with these fractures.
Major Features ^a^ The fracture is associated with minimal or no trauma, as in a fall from a standing height or less. The fracture originates at the lateral cortex and is substantially transverse in its orientation, although it may become oblique as it progresses medially across the femur. Complete fractures extend through both cortices and may be associated with a medial spike; incomplete fractures involve only the lateral cortex. The fracture is noncomminuted or minimally comminuted. Localized periosteal or endosteal thickening of the lateral cortex is present at the fracture site (“beaking” or “flaring”).
Minor Features Generalized increase in the cortical thickness of the femoral diaphysis. Unilateral or bilateral prodromal symptoms such as dull or aching pain in the groin or thigh. Bilateral incomplete or complete femoral diaphysis fractures. Delayed fracture healing.

ASBMR = American Society for Bone and Mineral Research; AFF = atypical femur fracture. ^a^ Excludes fractures of the femoral neck, intertrochanteric fractures with spiral subtrochanteric extension, periprosthetic fractures, and pathological fractures associated with primary or metastatic bone tumors and miscellaneous bone diseases (e.g., Paget’s disease, fibrous dysplasia).

## Data Availability

The data that support the findings of this study are openly available.

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
