# Peer review of "Bilateral Atypical Femoral Fractures after Bisphosphonate Treatment for Osteoporosis: A Literature Review"

_jcm, 2023, doi:10.3390/jcm12031038_

Round 1
Reviewer 1 Report
I think the review is correct. In fact, the description of the main characteristics in terms of duration of treatment, age of the patients, comorbidities and other risk factors for both unilateral and bilateral fractures have been known for a long time. The particularity would be the simultaneous non-sequential bilateral fractures. The authors correctly point out the limitations of referring few patients and that more studies are needed to evaluate these bilateral-simultaneous cases. It is important that the professionals who care for these patients and the patients themselves are aware of and sensitive to these infrequent but serious adverse effects in order to recognize them at an early stage.
Author Response
We deeply thank the reviewer for the detailed comments.

Reviewer 2 Report
Interesting topic and very thorough review of the literature. The review summarizes the main findings, diagnostic tools and pharmacological management of AFF. The case report is clearly expounded.
The methodological approach used is clear and well expounded, and the data collected are well illustrated in the tables.
The conclusions are in line with the data presented.
In section 4.4.2 at lines 380/387 it might be useful to mention this recent literature review PMID: 36290528.
Author Response
We thank the reviewers for their constructive comments to help us improve our manuscript. We have revised the manuscript according to reviewers’ comments.

Reviewer 3 Report
Anatomically, 47% of the fractures were in the near third, 51 % in the middle third, and 2 % in the far third of the femur.
91% of the operativly stabilized complete fractured femora, and 35% of the incomplete fractured femora underwent intramedullary nailing.
You should find out percentage of nailing with cephalomedullary component (trochanteric nail), and without (standard femoral nail), respectively. This question is of surgical interest in context with anatomical localization. Because fractures in the middle third can be reconstructed by intramedullary femoral shaft nails without cephalomedullary/trochanteric component. Furthermore, surgically important is the length of the used trochanteric nails, because there are standard and long versions.
Additionally, insert X-Ray of your case report, as the reader is informed about the length of the used nails.
Author Response

(The authors gave the same response as above.)
